# BEST-Route: Adaptive LLM Routing with Test-Time Optimal Compute

**Dujian Ding** [1]  **Ankur Mallick** [2]  **Shaokun Zhang** [3]  **Chi Wang** [4]  **Daniel Madrigal** [2]
**Mirian Del Carmen Hipolito Garcia** [2]  **Menglin Xia** [2]  **Laks V. S. Lakshmanan** [5]  **Qingyun Wu** [3,6]  **Victor Rühle** [2]

## Abstract

Large language models (LLMs) are powerful tools but are often expensive to deploy at scale. LLM query routing mitigates this by dynamically assigning queries to models of varying cost and quality to obtain a desired tradeoff. Prior query routing approaches generate only one response from the selected model and a single response from a small (inexpensive) model was often not good enough to beat a response from a large (expensive) model due to which they end up overusing the large model and missing out on potential cost savings. However, it is well known that for small models, generating multiple responses and selecting the best can enhance quality while remaining cheaper than a single large-model response. We leverage this idea to propose BEST-Route, a novel routing framework that chooses a model and the number of responses to sample from it based on query difficulty and the quality thresholds. Experiments on real-world datasets demonstrate that our method reduces costs by up to 60% with less than 1% performance drop.

## 1. Introduction

Large language models (LLMs) have revolutionized natural language processing (NLP) by delivering state-of-the-art performance across a wide range of tasks, from language understanding to creative writing, code generation, and beyond (Zhao et al., 2023). Their widespread deployment in applications like ChatGPT (OpenAI, a) and other conversational agents (Zheng et al., 2023; Zhang et al., 2024a;b) has made them a cornerstone of modern NLP systems. However, the superior performance of these models often comes with

substantial computational costs, driven by their large sizes and autoregressive text generation, making their deployment a challenge for both developers and users (Yu et al., 2022). The growing demand for LLM-backed services has spurred the development of innovative solutions to achieve efficiency without sacrificing quality.

The rising costs of LLM inference have spurred efforts to develop smaller, more cost-effective models such as self-consistency (Wang et al., 2023) and re-ranking (Chuang et al., 2023). However despite several innovations in this space (Dubey et al., 2024; Abdin et al., 2024), small models continue to come up short in terms of response quality when compared to the largest, most powerful models (see Figure 3 where y-axis measures response quality). Therefore an alternate line of work has focused on combining multiple models, small and large, to balance response quality and cost (Ding et al., 2024; Ong et al., 2024; Kim et al., 2023; Chen et al., 2023). Broadly speaking these works seek to leverage the small models to respond to easier queries while saving the large models for the more challenging queries thereby reducing costs without loss of response quality. In particular there are three sub-areas where this principle has been applied: 1) Query routing (e.g. Ong et al. (2024)) where a classifier/scorer rates the difficulty of an input and selects models accordingly, 2) Speculative decoding (e.g. Kim et al. (2023)) where a small (drafter) model returns candidate response tokens that are accepted/rejected by the large (verifier) model, and 3) Model cascades ( e.g. Chen et al. (2023)) where the query passes through the models sequentially, from the cheapest to the most costly, until either a satisfactory response is obtained or a pre-defined max number of models of the cascade is reached.

This work focuses on query routing and seeks to combine model selection with adaptive allocation of computing resources at test-time (Snell et al., 2024) to obtain notable response quality improvements compared to prior work while achieving substantial cost reduction. We observe that a major drawback of many prior query routing approaches is that they are unable to leverage extra compute and scale up the performance of the smaller (lower cost) models in their portfolio. Therefore they often end up routing all but the easiest queries to large models and thus provide little cost savings (this is shown in prior work (Ding et al., 2024) and

---

[1]University of British Columbia (work performed during internship at Microsoft) [2]Microsoft [3]Pennsylvania State University [4]Google (work performed while at Microsoft) [5]University of British Columbia [6]AG2AI, Inc.. Correspondence to: Dujian Ding <dujian.ding@gmail.com>.

*Proceedings of the 42nd International Conference on Machine Learning*, Vancouver, Canada. PMLR 267, 2025. Copyright 2025 by the author(s).

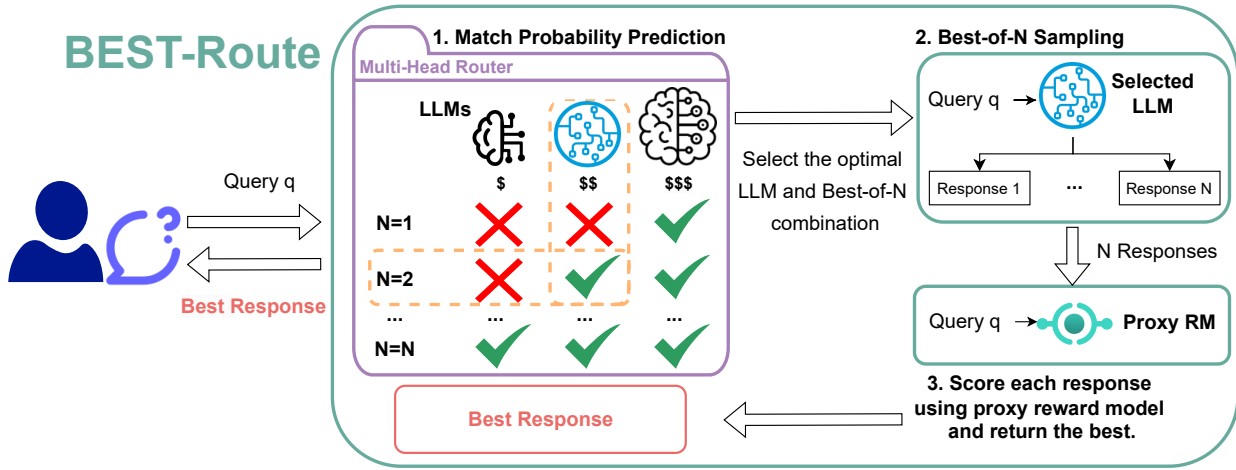

Figure 1. System overview of BEST-Route: **B**est-of-n **E**nhanced **S**ampling and **T**est-time **Route** Optimization.

in a couple of the baseline routing approaches in Figure 3).

We propose **B**est-of-n **E**nhanced **S**ampling and **T**est-time **Route** Optimization (**BEST-Route**), a novel LLM routing framework that effectively balances cost and quality through two key innovations. First, a cost-efficient multi-headed router dynamically assesses query difficulty to select the appropriate model and allocate computational resources. Second, a test-time optimal compute strategy leverages best-of-$n$ sampling (Stiennon et al., 2020; Nakano et al., 2021) to enhance small-model performance. This ensures that easy queries are routed to smaller, cheaper models with minimal sampling, while harder queries benefit from the advanced capabilities of larger models. Our framework employs a flexible model orchestration pipeline to adapt to varying cost-quality requirements. Specifically, our router predicts the likelihood that a small model, with best-of-$n$ sampling, can generate responses comparable to a powerful reference model (e.g., GPT-4o). This enables the selection of the optimal small-model and sampling strategy combination, ensuring high-quality responses at minimal cost. Unlike prior work with static sampling policies, our router adaptively determines the number of samples (and hence the compute allocation) needed for a small model to match the quality of the reference large model at the lowest cost. The overall routing framework is illustrated in Figure 1. Experiments on large-scale, real-world datasets (Section 5) demonstrate that our method achieves up to 60% cost reduction with less than 1% performance degradation, significantly improving upon prior routing techniques and contributing toward more efficient LLM service deployment.

The main contributions of this work are: 1) **Cost-efficient router design:** We propose a query difficulty-aware routing framework that allocates computational resources dynamically to achieve effective cost-accuracy trade-offs while adding minimal overhead. 2) **Test-time optimal compute strategy:** We introduce a best-of-$n$ sampling mechanism, allowing the router to select the most effective response which improves performance while still saving costs. 3) **Scalable real-world evaluations:** We demonstrate the effectiveness of our approach on large-scale datasets, achieving significant cost savings with minimal response quality drop.

Our work provides a robust solution for both LLM service providers and end-users, offering a flexible framework that balances cost and performance. By leveraging adaptive routing and test-time optimization, we advance the field of cost-efficient LLM inference, enabling broader accessibility and adoption of LLM-backed applications.

## 2. Related Work

**Efficient Machine Learning (ML) Inference.** Large language models (LLMs) have revolutionized natural language processing and related fields, offering remarkable effectiveness and generalizability. However, their increasing size comes at the cost of significant computational demands and prohibitive expenses for both training and deployment (Treviso et al., 2023; Bender et al., 2021). To address inference costs, prior research has focused on static efficiency optimizations such as model pruning (Hassibi et al., 1993; LeCun et al., 1989), quantization (Jacob et al., 2018; Vanhoucke et al., 2011), knowledge distillation (Hinton et al., 2015; Urban et al., 2016), and neural architecture search (Elsken et al., 2019; Zoph & Le, 2016). While these techniques produce smaller, lower-cost models, they offer fixed trade-offs between accuracy and efficiency, limiting their adaptability. Given that LLMs are expected to serve a wide range of tasks with varying accuracy and cost constraints, dynamic optimization approaches are essential to

enable more flexible and cost-effective inference (Ding et al., 2022; 2025).

**LLM Routing.** LLM routing has become an effective approach to provide dynamic optimization among multiple LLMs by striking good balances between overall response quality and incurred costs. In Ding et al. (2024); Ong et al. (2024), authors propose effective routing strategies to dynamically allocate queries between one strong-and-expensive LLM and one weak-yet-cheap LLM to reduce inference costs while maintaining high performance. Recent work extends the binary routing framework to accommodate a large set of LLMs. Srivatsa et al. (2024) investigates the feasibility of routing queries to the most suitable LLM from a selected set of models based on input features. FORC (Šakota et al., 2024) predicts the cost and performance of multiple LLMs using a meta-model and assigns queries to suitable models for optimized cost-performance trade-offs. ZOOTER (Lu et al., 2023) uses reward models to route queries to the most suitable LLMs, achieving high accuracy and reduced computational overhead. While effective, these routing approaches cannot utilize additional compute to enhance the performance of smaller, lower-cost models, particularly when the performance gap between models is substantial.

**Test Time Optimal Compute.** Test-time optimal compute techniques (Snell et al., 2024) such as best-of-$n$ sampling is effective for improving outcomes on challenging queries. These methods allow the model to explore multiple potential responses, increasing the likelihood of generating high-quality answers. In Brown et al. (2024), the authors observe that increasing the number of sampled responses boosts the probability of finding correct solutions for hard queries, especially in tasks like coding and mathematics. Similarly, Chen et al. (2024a) finds out that increasing the number of LLM calls improves performance on "easy" queries and highlights the importance of adapting compute strategies to query difficulty. More recently, Gui et al. (2024); Jinnai et al. (2024) demonstrates the effectiveness of best-of-$n$ sampling for aligning LLM outputs to human preferences by selecting the best response among multiple samples. However, this line of work primarily focuses on scaling the test-time compute of a *single* LLM, missing the opportunity of harnessing the respective strengths of multiple models.

**Other Multi-LLM Inference Techniques.** Speculative decoding (Leviathan et al., 2023; Kim et al., 2023; Chen et al., 2024b; Narasimhan et al., 2024) speeds up decoding of expensive LLMs by invoking small decoders on the "easy" decoding steps. Unlike LLM routing, which optimizes query traffic distribution among multiple LLMs to balance cost and performance, speculative decoding solely focuses on accelerating the decoding process within a single expensive model by mitigating the inefficiencies of auto-regressive

text generation. Model Cascades (Chen et al., 2023; Gupta et al., 2024; Yue et al., 2023) performs inference by sequentially calling LLMs with effective post-hoc deferral rules based on either the confidence scores or answer consistency of weaker LLMs. More recently, a line of work studies how to combine the capacity from different LLMs to further improve response quality. Mixture-of-Agents (Wang et al., 2024b) leverages the collective strengths of multiple LLMs by introducing a layered architecture where agents iteratively refine responses. PackLLM (Mavromatis et al., 2024) introduces a test-time fusion approach that minimizes perplexity to determine the contribution of each model in a weighted ensemble. However, these approaches typically call more than one LLM for a single query and can incur significant computational overheads.

## 3. Problem Formulation

### 3.1. Motivation

**Varying Query Difficulty.** Queries naturally vary in difficulty according to their complexity, ambiguity, and task requirements. For example, a query like "*Rewrite the sentence so that it's in the present tense – 'She had worked at the company for the past 3 years'.*" is straightforward and can be accurately resolved by a smaller or less capable model. In contrast, a more complex query like "*Can you summarize the implications of quantum entanglement on secure communication?*" requires nuanced understanding and reasoning, demanding the capabilities of a larger, more powerful model or additional test-time compute resources such as sampling multiple responses. In Ding et al. (2024); Ong et al. (2024), authors demonstrate that complex or ambiguous tasks benefit from the broader knowledge and reasoning capabilities of larger models, and therefore difficult queries can be routed to larger and more capable LLMs to maintain response quality, while easy queries can be served by smaller LLMs to achieve significant cost savings.

**Sub-optimal Model Utilization.** While several works have explored leveraging query difficulty variation by routing queries to appropriate models, they face limitations that prevent them from fully utilizing available LLM infrastructure for maximum gains. Many approaches focus solely on routing between two models—one small and one large (Kag et al., 2022; Ding et al., 2024; Ong et al., 2024). This is a reasonable starting point, as the binary case is easier to analyze, and early LLM inference platforms offered only a limited selection of models. However, with platforms like Hugging Face (HuggingFace) now providing a diverse range of LLMs across the cost-quality spectrum, routing across all available models is crucial for achieving the best trade-off.

While some works (Šakota et al., 2024; Shnitzer et al., 2023; Srivatsa et al., 2024) attempt to route among more than

two models, they often fail to fully utilize smaller models, frequently defaulting to querying the largest model. Our experiments in Section 5 confirm this trend. A common technique for enhancing small-model response quality is best-of-$n$ sampling, where multiple responses are generated, and the best one is selected (Stiennon et al., 2020; Nakano et al., 2021). However, prior approaches are often too costly and time-consuming for real-time inference due to the extensive usage of large (LLM-based) reward models (Lambert et al., 2024). To address this, we first develop a low-cost best-of-$n$ sampling method to enhance small-model response quality at inference time. We then train a router to select the optimal model and number of responses, achieving the best quality at the lowest cost.

### 3.2. Problem Setting

We propose a routing framework for efficiently serving user queries using multiple large language models (LLMs) with varying cost and quality, such as those offered by popular LLM serving platforms (HuggingFace; OpenAI, b). Our system consists of a powerful reference model, $M_{\text{ref}}$ (e.g., GPT-4o), and a set of smaller, more cost-efficient models, $\mathcal{M}$. Given a query $q$, we can directly return one response from $M_{\text{ref}}$ or return the best-of-$n$ responses from a model in $\mathcal{M}$. A router must efficiently select the model and sample count to minimize inference costs while preserving response quality near that of $M_{\text{ref}}$, with minimal added latency/cost overheads. It is worth noting that we always route each query to a single LLM during inference rather than employing an ensemble (Jiang et al., 2023) or a cascade (Chen et al., 2023), both of which involve multiple LLM calls per query and can lead to substantial computational overheads.

### 3.3. Evaluation Metric

**Response Quality.** Automatically evaluating text generation remains a challenging and extensively researched problem. Traditional metrics like BLEU and ROUGE, originally developed for machine translation and summarization, often exhibit weak alignment with human judgment and have limited applicability across diverse NLP tasks (Blagec et al., 2022). Recent studies (Jiang et al., 2023; Wang et al., 2024a) suggest that LLMs, when properly prompted or fine-tuned, can provide more human-aligned evaluations. In this work, we adopt armoRM (Wang et al., 2024a), a fine-tuned Llama3-8B model, to assess response quality. armoRM ranks highly on advanced evaluation model benchmarks (Lambert et al., 2024) and, due to its relatively small size (8B), enables feasible large-scale evaluation. We also independently demonstrate the effectiveness of armoRM scores in Appendix C.

**Inference Cost.** The cost of running a model can be measured using various metrics, such as FLOPs, latency, or mon-

etary expenses. While FLOPs offer a hardware-independent measure, they do not always correlate well with practical concerns like wall-clock latency, energy usage, or financial costs, which are more relevant to end users (Dao et al., 2022). For LLM service users, inference costs primarily consist of input and output token expenses, calculated by multiplying the respective token counts by their unit prices (see Table 6). In this work, we quantify inference costs in USD and also report the latency of our routing framework as part of our evaluation.

## 4. Routing Framework

We first develop a memory efficient approach for best-of-$n$ sampling from LLMs and then design a router that selects the appropriate LLM and number of samples for each query.

### 4.1. Memory Efficient Best-of-$n$ Sampling

Best-of-$n$ sampling (Stiennon et al., 2020; Nakano et al., 2021; Gui et al., 2024; Brown et al., 2024) enhances LLM response quality by generating $n$ candidates and selecting the best, leveraging output variability to better align with quality expectations. A straightforward option for best-of-$n$ sampling is to score each response using human evaluators or LLM-as-a-judge systems (Zheng et al., 2023), but integrating human scorers is impractical for real-time inference, and LLM-based scoring adds substantial compute and memory costs. Instead, we use a smaller proxy reward model to approximate these costly scoring methods and efficiently select the best response.

Given an input query $q$, and $n$ responses $s_1(q), s_2(q), \ldots, s_n(q)$ obtained from the same LLM, let $R_{\text{GT}}(q, s(q))$ denote the *ground truth* quality score of a response $s$, obtained via an expensive scoring approach such as human or LLM-as-a-judge scoring and let $R_{\text{proxy}}(q, s(q))$ denote the score from the proxy reward model. We will use the notation $R_{\text{GT}}(s(q)), R_{\text{proxy}}(s(q))$ for brevity.

We aim to select *the best* out of $n$ responses as per the ground truth reward $R_{\text{GT}}(s(q))$. Thus, if the ordering of responses is preserved under the proxy model i.e. $R_{\text{proxy}}(s_i(q)) > R_{\text{proxy}}(s_j(q))$ whenever $R_{\text{GT}}(s_i(q)) > R_{\text{GT}}(s_j(q))$ then it can be used for best-of-$n$ sampling.

Since we only want the proxy model to preserve the ranking of responses, we train $R_{\text{proxy}}$ by minimizing a pairwise ranking loss on a set $\mathcal{P}$ of training pairs constructed as $\mathcal{P} = \{(s, s') | R_{\text{GT}}(s) > R_{\text{GT}}(s')\}$. The loss function is

$$\mathcal{L}_{\text{rank}} = -\frac{1}{|\mathcal{P}|} \sum_{(s,s') \in \mathcal{P}} \log \sigma \left( R_{\text{proxy}}(s) - R_{\text{proxy}}(s') \right), \quad (1)$$

where $\sigma(x) = \frac{1}{1+e^{-x}}$ is the sigmoid function. The loss formulation is obtained from previous reward modeling work (Ouyang et al., 2022).

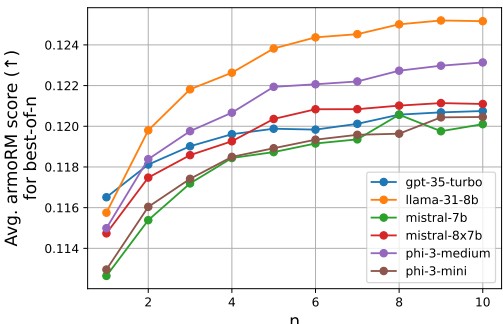

*Figure 2.* armoRM score of response selected through best-of-$n$ sampling using our proxy reward model consistently increases.

To construct the training set $\mathcal{P}$, we generate $n = 20$ sample responses $\mathcal{S} = \{s_1(q), s_2(q), \ldots, s_{20}(q)\}$ for each training query $q \in \mathcal{Q}$ and compute $R_{\text{GT}}(s(q))$ using the armoRM score (Wang et al., 2024a). While armoRM ranks highly on benchmarks like RewardBench (Lambert et al., 2024), our framework supports any LLM or human judge as ground truth. Next, we select three responses per query: **worst** ($s_{\text{worst}}$), **median** ($s_{\text{median}}$), and **best** ($s_{\text{best}}$), and form two pairs: $(s_{\text{worst}}, s_{\text{median}})$ and $(s_{\text{median}}, s_{\text{best}})$. We obtain such paris for all queries and aggregate them to form $\mathcal{P}$ which is used to train $R_{\text{proxy}}$ by minimising Equation (1).

Note that $n$ responses from each query can generate $\binom{n}{2}$ pairs but many pairs may have similar quality scores, making fine-grained ranking difficult and hindering training performance. We use only the worst, median, and best responses to prioritize pairs with the largest ground truth score differences. Since best-of-$n$ sampling focuses on selecting the highest-quality response, minor misclassifications among similar-quality pairs has minimal impact. Our approach ensures that the proxy reward model effectively guides high-quality sampling while reducing training complexity.

During inference, for a given query $q$, we (1) generate $n$ sample responses $\mathcal{S} = \{s_1(q), s_2(q), \ldots, s_n(q)\}$, (2) compute the proxy scores for each sample: $R_{\text{proxy}}(s_i(q))$ for $i = 1, 2, \ldots, n$, and (3) select the response with the highest proxy score $s^* = \arg\max_{s \in \mathcal{S}} R_{\text{proxy}}(s)$. Figure 2 plots the average armoRM score ($R_{\text{GT}}$) for the best-of-$n$ responses selected by our proxy reward model for the test set (see Section 5) with varying $n$. The consistent increase in armoRM score as $n$ increases shows that $R_{\text{proxy}}$ works as expected. In particular it is not seen to suffer from reward hacking (Skalse et al., 2022) (i.e. poor correlation with $R_{\text{GT}}$) often seen when optimizing imperfect proxy rewards.

### 4.2. Test-time Optimal LLM Routing

Recall from Section 3.2 that the goal of routing is to select either a powerful reference model $M_{\text{ref}}$ (e.g., GPT-4o) and return a single response from it, or select a smaller model

---

**Algorithm 1** BEST-Route

**Input:** Query $q$, Maximal sample number $n$, Match probability threshold $t$, Proxy reward model $R_{\text{proxy}}$; Models $\{M_1, \ldots, M_K\}$, Reference model $M_{\text{ref}}$; Average output lengths avg_output_length$[M]$; Input and output token prices input_token_price$[M]$, output_token_price$[M]$;
**Output:** Final response.

/* 1.   Compute Match Probabilities:   */
**foreach** $M \in \{M_1, \ldots, M_K\}$ **do**
  **for** $i = 1$ **to** $n$ **do**
    match_prob$[(M, i)]$ $\leftarrow$
      MultiHeadRouter.predict_match_prob$(q, M, i, M_{\text{ref}})$

/* 2.   Filter and Compute Costs:   */
valid_comb $\leftarrow \emptyset$
**foreach** $(M, i) \in$ *match_prob* **do**
  **if** *match_prob*$[(M, i)] \geq t$ **then**
    costs$[(M, i)]$ $\leftarrow$ $i \times$ avg_output_length$[M]$ $\times$ output_token_price$[M]$ $+$ $q$.input_length $\times$ input_token_price$[M]$
    valid_comb $\leftarrow$ valid_comb $\cup \{(M, i)\}$

/* 3.   Select Optimal Combination:   */
**if** *valid_comb* $\neq \emptyset$ **then**
  $(M^*, i^*) \leftarrow \arg\min_{(M,i) \in \text{valid\_comb}}$ costs$[(M, i)]$
**else**
  $(M^*, i^*) \leftarrow (M_{\text{ref}}, 1)$

/* 4.   Execute Sampling Strategy:   */
Draw $i^*$ samples $\{s_1, \ldots, s_{i^*}\}$ from $M^*$ for query $q$ and compute $s^*_{\text{small}} \leftarrow \arg\max_{s \in \{s_1, \ldots, s_{i^*}\}} R_{\text{proxy}}(s)$
**return** $s^*_{\text{small}}$

---

from the set $\mathcal{M}$ and return the best-of-$n$ responses from it using the sampling approach described above. The key intuition here is that *for many small models, sampling multiple responses and selecting the best is often still cheaper than sampling a single response from the reference model* (see Section 5 for cost breakups). We will describe the router design by first introducing a pair-wise router for routing between two models, then extending it to a matrix-of-routers that can route between more than two models, and finally describing our multi-headed router which is a *single* router that approximates the matrix-of-routers while signficantly reducing cost/latency overheads.

**Pair-Wise Router.** Given a query $q$ and two candidate inference options—a large reference model $M_{\text{ref}}$ (e.g., GPT-4o) and a smaller model $M_{\text{small}}$ (e.g., Llama-3.1-8b), and a specific value of $n$, we can augment $M_{\text{small}}$ with best-of-$n$ sampling by generating $n$ samples and then selecting the best using our proxy reward model as,

$$s^*_{\text{small}} = \arg\max_{s \in \mathcal{S}} R_{\text{proxy}}(s), \quad \mathcal{S} = \{s_1(q), s_2(q), \ldots, s_n(q)\},$$
$$(2)$$

We want our router to estimate the likelihood of $s_{\text{small}}^*$ being at least as good as $s_{\text{ref}}$, the response from $M_{\text{ref}}$, under the *ground-truth* reward, $R_{\text{GT}}$. Therefore for each training query $q$, we generate a label

$$y_n(q) = Pr[R_{\text{GT}}(s_{\text{small}}^*) \geq R_{\text{GT}}(s_{\text{ref}})] \quad (3)$$

Our pair-wise router is trained to minimize the cross entropy loss,

$$\mathcal{L}_{\text{pair}} = -\frac{1}{|\mathcal{Q}|} \sum_{q \in \mathcal{Q}} \left( y_n(q) \log p_n(q) + (1 - y_n(q)) \log(1 - p_n(q)) \right),$$
$$(4)$$

where $p_n(q)$ is the probability predicted by the router that the best-of-$n$ response, from $M_{\text{small}}$ is as good as a single response from $M_{\text{ref}}$ for that value of $n$, termed as *match probability*.

**Matrix-of-Routers.** Let there be $K$ small models in the set $\mathcal{M}$. We can train $K \times N$ distinct pair-wise routers, in our matrix-of-routers. Each router predicts the match probability between a smaller model with best-of-$n$ sampling and $M_{\text{ref}}$ for a specific value of $n$ ($1 \leq n \leq N$).

**Cost-Efficient Multi-Head Router.** Training and deploying $K \times N$ separate pair-wise routers is computationally expensive. To address this, we propose a cost-efficient **multi-head router** design.

Specifically, we leverage a shared BERT-style backbone Router$_{\text{shared}}$ encodes the query $q$ into a shared representation $\mathbf{h}_q$ and train $K \times N$ lightweight classification heads Head$_{k,n}$ separately to predict:

$$p_{k,n}(q) = \sigma\left(\mathbf{w}_{k,n}^\top \mathbf{h}_q + b_{k,n}\right) \ 1 \leq k \leq K, \ 1 \leq n \leq N \quad (5)$$

where $\sigma(x) = \frac{1}{1+e^{-x}}$ is the sigmoid function, $\mathbf{w}_{k,n}$ is the weight vector, $b_{k,n}$ is the bias term, and $p_{k,n}(q)$ denotes the probability that the best-of-$n$ response, from the $k^{\text{th}}$ model in $\mathcal{M}$ is as good as a single response from $M_{\text{ref}}$ for that $n$.

At inference time, users can set thresholds on $p_{k,n}(q)$ to balance cost and accuracy. Higher thresholds favor the reference model, improving quality at increased costs. Multiple small-model and best-of-$n$ combinations can meet a given threshold. BEST-Route effectively selects among them using cost estimation to ensure high-quality responses at minimal cost.

Specifically, total cost comprises prompt and response costs, computed as the product of token count and unit token price for inputs and outputs, respectively. Since output length is unknown at inference, we estimate it using average training data lengths. We demonstrate that this estimation has low error and can effectively support the development of efficient routing frameworks (see Appendix B.1).

The overall test-time optimal LLM routing framework is as depicted in Algorithm 1. We first use the Multi-Head Router to predict the match probability for each model and

best-of-$n$ sampling strategy against the reference model. Secondly, we identify combinations where the predicted match probability meets or exceeds the threshold and compute the incurred cost for each valid combination [1]. Next, from the valid combinations, we select the one with the smallest estimated cost. If no combination satisfies the threshold, we use the reference model with one single call. Lastly, for the selected model and sampling strategy, we draw the desired number of samples, evaluate them using the proxy reward model, and return the response with the highest proxy score.

## 5. Evaluation

### 5.1. Evaluation Setup

**Dataset.** We introduce a large-scale dataset covering diverse tasks, including question answering, coding, and safety evaluation, with examples collected from multiple sources (see Appendix A.1). The dataset consists of 10K instruction examples, split into 8K/1K/1K for training, validation, and testing. We evaluate BEST-Route across 8 popular LLMs—GPT-4o, GPT-3.5-turbo, Llama-3.1-8B, Mistral-7B, Mistral-8x7B, Phi-3-mini, Phi-3-medium, and Codestral-22B—by generating 20 responses per example. We further perform out-of-distribution (OOD) evaluation of BEST-Route using MT-Bench (Zheng et al., 2023).

**Router and Proxy Reward Model.** We use DeBERTa-v3-small (He et al., 2020) (44M) as the backbone to train our Multi-Head Router, while the proxy reward model is fine-tuned from OpenAssistant RM [2], a DeBERTa-v3-large model (300M). We train both Multi-Head Router and the proxy reward model with the corresponding loss from Section 4 for 5 epochs and use the validation set to choose the best checkpoints for final evaluation. All inference experiments are conducted using paid API access from OpenAI [3], AzureML [4], and Mistral AI [5], while router training and inference are performed on an NVIDIA A100 GPU (80GB RAM). Codes will be released upon acceptance of this work.

**Evaluation Metrics.** We assess response quality using armoRM scores (Wang et al., 2024a) and measure efficiency based on incurred inference costs, which include input and output token pricing (see Table 6). We also report trained router performance using BLEU and ROUGE in Section 5.5.

---

[1] Input tokens are only charged once because most modern LLMs support returning multiple responses at once for a given query. For example, you can set the "num_return_sequences" hyperparameter for HuggingFace LLMs to tune the number of independently computed returned sequences for each query.

[2] https://huggingface.co/OpenAssistant/reward-model-deberta-v3-large-v2

[3] https://openai.com/api/pricing/

[4] https://azure.microsoft.com/en-us/pricing/details/phi-3/

[5] https://mistral.ai/technology/#pricing

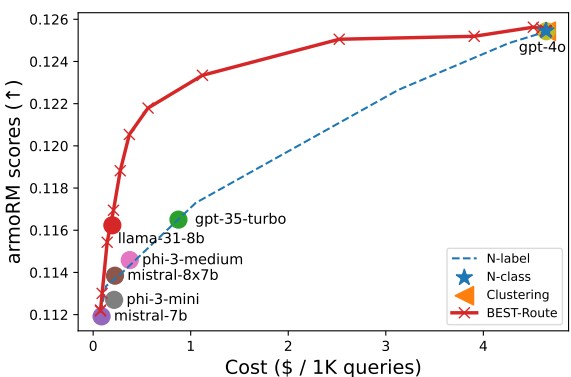

*Figure 3.* Routing performance results.

| Cost Reduction (%) | Response Quality Drop (armoRM score) w.r.t. always using GPT-4o (%) | |
|---|---|---|
| | N-label | BEST-Route |
| 10 | 0.63 | **0.19** |
| 20 | 1.17 | **0.21** |
| 40 | 3.26 | **0.47** |
| 60 | 5.08 | **0.80** |
| N-class | 0.07% cost reduction with 0% quality drop. | |
| Clustering | 0% cost reduction with 0% quality drop. | |

*Table 1.* Cost reduction v.s. performance drops. Performance drops are computed w.r.t. always using the reference model (GPT-4o).

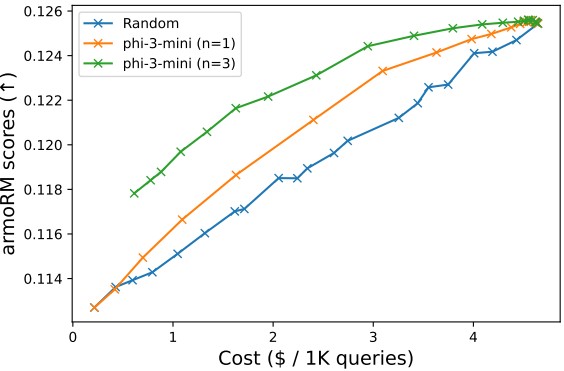

*Figure 4.* Routing performance between GPT-4o and Phi-3-mini with best-of-$n$ sampling where $n = 1, 3$.

**Baselines.** We compare BEST-Route against 3 routing baselines from prior work (Srivatsa et al., 2024), including (1) **N-class Routing** – a BERT-based router aiming to predict the best LLM for a given input query, (2) **N-label Routing** – a BERT-based router predicting all capable LLMs and selecting the cheapest one, (3) **Clustering-based Routing** – fitting K-Means clustering model to query-specific features and routing queries to the optimal LLM corresponding to their assigned cluster. We further consider **Model Cascade** baselines (Yue et al., 2023) and report results in Appendix B.2. All baseline details are provided in Appendix A.2.

**Experiments.** We investigate our test-time optimal LLM routing framework. We evaluate the routing performance in Section 5.2 (Figure 3 and Table 1), validate that the router is indeed adaptively distributing queries between different LLMs to achieve good cost-v.s.-accuracy trade-offs in Section 5.3, demonstrate that our routing framework is of negligible compute overhead in Section 5.4, examine the router generalizability in Section 5.5, show that our cost estimation is of low estimation error in Appendix B.1, and present more performance results compared to model cascade baselines in Appendix B.2. Our code is available at https://github.com/microsoft/best-route-llm.

### 5.2. Router Performance Results

We evaluate the effectiveness of routing queries across LLMs with significant performance gaps (Figure 3), with numerical results summarized in Table 1. Routing is inherently challenging as the reference model (e.g., GPT-4o) dominates for most queries, making cost reduction difficult without sacrificing quality.

Unlike BEST-Route, which enables adaptive cost-accuracy trade-offs through a tunable threshold, N-class Routing and Clustering-based Routing make fixed routing decisions, offering no flexibility. As a result, they largely default to using the reference model for nearly all queries, achieving

minimal cost savings. Similarly, N-label Routing struggles to reduce costs while preserving response quality, leading to over 5% performance drop at 60% cost reduction.

In contrast, BEST-Route consistently outperforms all baselines, achieving higher cost reductions with lower performance degradation. Notably, BEST-Route achieves 60% cost reduction with only a 0.8% quality drop, up to 4.28% better than all baselines (Table 1).

We also examine the impact of best-of-$n$ sampling (Figure 4) by comparing GPT-4o and Phi-3-mini at $n = 1, 3$. The results show that best-of-$n$ sampling significantly enhances routing performance, achieving better cost-accuracy trade-offs. We further compare BEST-Route with best-of-$n$ sampling for each LLM (Figure 5). Best-of-$n$ offers fixed trade-offs for each model and $n$ pair and often yields lower-quality responses (e.g., 4.9% quality drop for Phi-3-mini, 1.1% for LLaMA-3.1-8B at $n = 5$). In contrast, BEST-Route offers flexible trade-offs, achieving 20% cost reduction with only 0.21% quality drop, and 40% cost reduction with 0.47% drop, with the max sampling number $n = 5$.

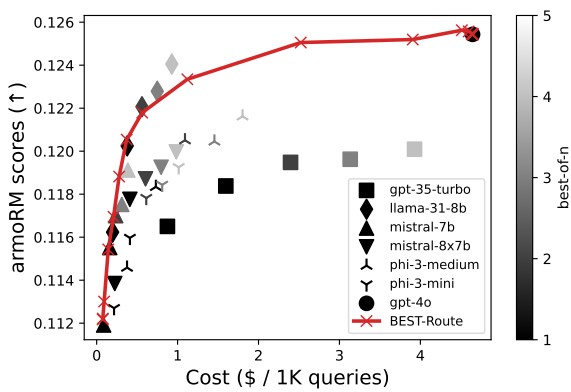

Figure 5. BEST-Route v.s. best-of-$n$ for each single LLM.

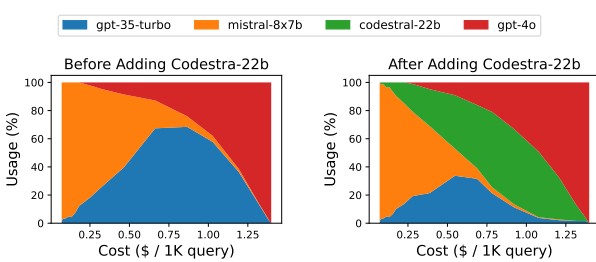

Figure 6. Model usage before and after adding Codestral-22b on coding queries.

### 5.3. Router Validation Results

We validate that BEST-Route effectively balances cost and accuracy by adaptively routing queries between small and large models and leveraging specialized models for further gains. We conduct experiments on coding queries with GPT-4o, GPT-3.5-turbo, Mistral-8x7b, and the specialized coding model Codestral-22b, analyzing cost-performance trade-offs and traffic distribution before and after adding Codestral-22b.

As shown in Figure 6, when Codestral-22b is absent, BEST-

| Cost Reduction (%) | Response Quality Drop (armoRM score) w.r.t. always using GPT-4o (%) | |
|---|---|---|
| | Before Adding Codestral-22b | After Adding Codestral-22b |
| 10 | 0.50 | **-0.10** |
| 20 | 0.77 | **-0.50** |
| 40 | 2.67 | **2.49** |
| 60 | 5.64 | **5.11** |

Table 2. Cost reduction v.s. performance drops on coding queries. Performance drops are computed w.r.t. always using the reference model (GPT-4o).

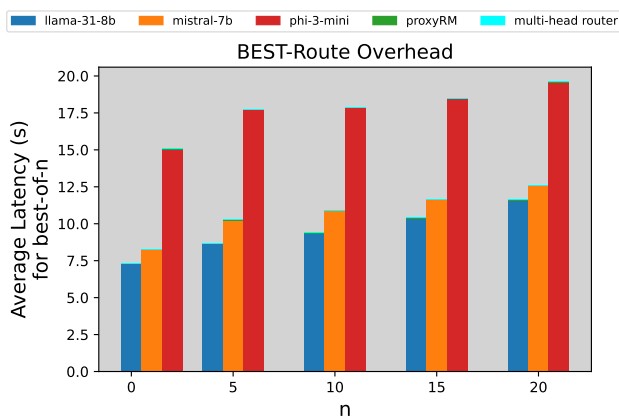

Figure 7. Overhead analysis.

| Cost Reduction (%) | Response Quality Drop (armoRM score) w.r.t. always using GPT-4o (%) | |
|---|---|---|
| | N-label | BEST-Route |
| 10 | 0.88 | **0.25** |
| 20 | 2.29 | **0.43** |
| 40 | 4.41 | **1.56** |
| 60 | 5.89 | **1.59** |
| N-class Clustering | 0% cost reduction with 0% quality drop. | |
| | 0% cost reduction with 0% quality drop. | |

Table 3. OOD evaluation on MT-Bench. Performance drops are computed w.r.t. always using the reference model (i.e., GPT-4o).

Route primarily selects Mistral-8x7b under strict cost constraints due to its low cost and reasonable accuracy (see Table 6 in the Appendix). As the budget increases, queries shift towards GPT-3.5-turbo and GPT-4o for improved accuracy. However, after Codestral-22b is added, a significant portion of coding queries is redirected from GPT-3.5-turbo to the specialized model, leading to better cost-performance trade-offs (Table 2). Notably, BEST-Route achieves up to 20% cost reduction while *exceeding* GPT-4o performance (negative response quality drop in Table 2 corresponds to response quality *gain* over always using GPT-4o). This suggests that query routing can not only save cost but also improve performance, consistent with prior findings for the two-model case (Ding et al., 2024).

### 5.4. Router Latency

We measure the latency of BEST-Route and compare it with the inference latency of different LLMs. We locally deploy Llama-3.1-8b, Mistral-7b, and Phi-3-mini that we use in our experiments to generate responses to user queries for evaluation purpose. We do not measure the latency of LLM APIs (e.g., GPT-3.5-turbo) because they introduce

| Cost Reduction (%) | Response Quality Drop w.r.t. always using GPT-4o (%) | | | |
|---|---|---|---|---|
| | BLEU | | ROUGE | |
| | N-label | BEST-Route | N-label | BEST-Route |
| 10 | 6.57 | **3.61** | 6.10 | **3.88** |
| 20 | 13.13 | **6.07** | 11.65 | **7.27** |
| 40 | 25.80 | **12.76** | 22.26 | **15.78** |
| 60 | 31.70 | **18.07** | 27.62 | **21.97** |
| N-class Clustering | 0.2% cost reduction with 0.7% quality drop. 0% cost reduction with 0% quality drop. | | 0.2% cost reduction with 0.9% quality drop. 0% cost reduction with 0% quality drop. | |

*Table 4.* Routing performance under BLEU and ROUGE. Performance drops are computed w.r.t. always using the reference model (i.e., GPT-4o).

additional delays due to network latency and queuing, and inference latency is expected to be significantly higher than that of the router.

The latency of BEST-Route primarily stems from three components: (1) match probability prediction by the Multi-Head Router, (2) LLM generation latency for producing $n$ responses, and (3) best-of-$n$ sampling overhead from using the proxy reward model. As shown in Figure 7, the routing overhead is negligible compared to LLM inference time. For instance, at $n = 20$, match probability prediction takes 0.04s and best-of-$n$ sampling adds 0.58s, making the total overhead 18.7× faster than the fastest LLM (Llama-3.1-8b). Moreover, increasing $n$ has only a marginal impact on overall latency. As $n$ grows from 1 to 20, LLM generation latency increases by just 30% for Phi-3-mini, 53.7% for Mistral-7b, and 59.3% for Llama-3.1-8b, demonstrating the efficiency of our best-of-$n$ sampling strategy.

### 5.5. Router Generalizability

To investigate the generalizability of BEST-Route, we evaluate the trained routers on the out-of-distribution (OOD) dataset – MT-Bench (Zheng et al., 2023), and more metrics (e.g., BLEU and ROUGE) in addition to armoRM scores.

As shown in Table 3, BEST-Route consistently outperforms all baselines on MT-Bench. For example, it achieves 60% cost reduction with only a 1.59% performance drop – up to 4.3% better than the strongest baseline. In contrast, N-class and clustering-based routing often default to using GPT-4o, yielding minimal cost savings, while N-label routing suffers notable quality drops especially at high cost reduction rates. Similarly, as shown in Table 4, we observe that BEST-Route consistently achieves better trade-offs than all baselines under both BLEU and ROUGE (e.g., N-label routing has up to 31.7% BLEU drop at 60% cost reduction, vs. only 18.07% drop for BEST-Route). These results demonstrate the robustness of BEST-Route under distribution shifts and generalizability to alternative quality metrics.

## 6. Limitations

While BEST-Route effectively reduces inference costs while maintaining high response quality, our approach has some limitations that warrant further investigation:

**Dependency on Proxy Reward Model Accuracy.** Our best-of-$n$ sampling strategy relies on the proxy reward model to rank generated responses effectively. Although our experiments demonstrate strong alignment between the proxy model and ground-truth evaluations, potential misalignment in certain cases may result in suboptimal response selection.

**Scalability to Extremely Large Model Pools.** While BEST-Route extends routing beyond binary selection to a diverse set of models, its effectiveness in handling extremely large model pools (e.g., hundreds of LLMs) remains unexplored. Efficiently scaling our router design to such a vast space may require additional optimizations.

## 7. Conclusion

In this work, we introduced BEST-Route, a novel framework for adaptive LLM routing that optimizes inference costs while maintaining high response quality. Our approach combines a cost-efficient routing strategy with test-time optimal compute through best-of-$n$ sampling, enabling dynamic model selection tailored to query difficulty. Through extensive evaluations on real-world datasets, we demonstrate that BEST-Route achieves up to 60% cost reduction with less than 1% performance drop, significantly outperforming prior routing frameworks. Our multi-head router design allows for fine-grained trade-offs between accuracy and efficiency, while our cost-aware best-of-$n$ sampling strategy further enhances response quality without unnecessary computational overhead. Our findings suggest that BEST-Route provides a flexible and effective solution for cost-efficient LLM inference, paving the way for more accessible and adaptive LLM services.

## Acknowledgements

The authors would like to thank Yizhu Jiao, Xuefeng Du, Hao Kang, Yinfang Chen, Ruomeng Ding, and Wenyue Hua for helpful discussions.

## Impact Statement

This paper presents work whose goal is to advance the field of Machine Learning. There are many potential societal consequences of our work, none which we feel must be specifically highlighted here.

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

| Scenario | Source | # Examples |
|---|---|---|
| Question Answering | MixInstruct | 6,000 |
| Coding | RewardBench | 984 |
| **2K total** | CodeUltraFeedback | 1,016 |
| Safety | RewardBench | 740 |
| **2K total** | BeaverTails | 1,260 |
| Total | Mix | 10,000 |

*Table 5.* Dataset statistics. It contains 10K examples and we randomly split the dataset into train/dev/test in 8K/1K/1K sizes.

| Model | Inputs ($/1M Tokens) | Outputs ($/1M Tokens) |
|---|---|---|
| GPT-4o | 5 | 15 |
| GPT-3.5-turbo | 3 | 6 |
| Llama-3.1-8b | 0.3 | 0.61 |
| Mistral-7b | 0.25 | 0.25 |
| Mistral-8x7b | 0.7 | 0.7 |
| Phi-3-mini | 0.3 | 0.9 |
| Phi-3-medium | 0.5 | 1.5 |
| Codestra-22b | 1 | 3 |

*Table 6.* LLM input and output token prices.

# A. Experiment Details

## A.1. Dataset

We introduce a new dataset to evaluate the effectiveness of different routing strategies across a wide range of tasks (e.g., question answering, coding, safety evaluation). We collect a large-scale set of instruction examples primarily from four sources, as shown in Table 5. The broad range of tasks in the dataset enables us to train a generic routing framework that will be effective across different scenarios. We sample 8K examples for training, 1K for validation, and 1K for testing. We then run $K = 8$ popular LLMs – GPT-4o, GPT-3.5-turbo, Llama-3.1-8b, Mistral-7b, Mistral-8x7b, Phi-3-mini, Phi-3-medium, and a specilized coding model Codestral-22b – to generate 20 responses on these 10k examples.

## A.2. Baselines

We consider three baselines from previous LLM routing work in the main evaluation section.

1. **N-class Routing**. A BERT-based router aiming to predict the best LLM for a given input query.

2. **N-label Routing** (Srivatsa et al., 2024). Similarly, a BERT-based router aiming to predict all LLMs capable for a given input query and selecting the cheapest LLM if there are multiple candidates.

3. **Clustering-based Routing** (Srivatsa et al., 2024). We apply a K-Means clustering model to query-specific features extracted from the training data using a TF-IDF vectorizer (Bafna et al., 2016) to identify discrete clusters. For each cluster in the training set, the most effective LLM is selected. During inference, test set queries are routed to the optimal LLM corresponding to their assigned cluster. We choose $K = 50$ by default as suggested in (Srivatsa et al., 2024).

We further compare BEST-Route with **Model Cascades** (Yue et al., 2023) to further demonstrate the effectiveness of our approach. Results are summarized in Appendix B.2. Specifically, **Model Cascades** ranks all LLMs based on their average inference costs on the training data, which are calculated as the sum of the prompt cost and the average response cost. For each LLM in the cascade, we sequentially sample $K$ responses and stop once the most consistent response $i^*$ achieves a consistency score above a predefined threshold. The consistency score for a response $i \in [K]$ is defined as the average of

| Model | Estimation Error ($ / query) |
|---|---|
| GPT-4o | 0.0027 |
| GPT-3.5-turbo | 0.0006 |
| Llama-3.1-8b | 0.0001 |
| Mistral-7b | 0.0001 |
| Mistral-8x7b | 0.0001 |
| Phi-3-mini | 0.0002 |
| Phi-3-medium | 0.0003 |
| Codestra-22b | 0.0004 |

*Table 7.* LLM response cost estimation error.

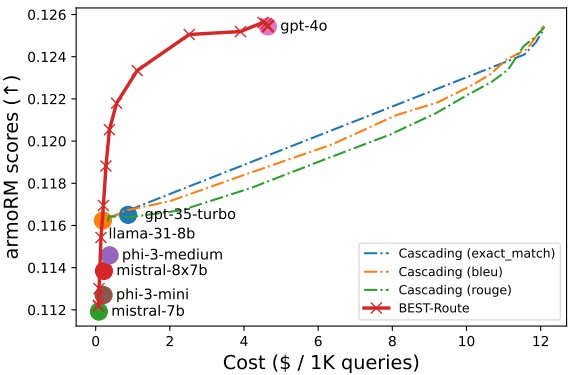

*Figure 8.* Routing performance results compared to model cascades baselines.

the agreement function values between $i$ and $j \in [K]$, expressed as

$$\text{consistency}(i) = \frac{1}{K} \sum_{j \in [K]} \text{agree\_func}(i, j).$$

Following (Yue et al., 2023), we set $K = 5$ and measure consistency using three agreement functions: exact_match, BLEU (Papineni et al., 2002), and ROUGE (Lin, 2004) scores. The most consistent response, $i^*$, is determined as

$$i^* := \arg\max_i \text{consistency}(i) \text{ for } i \in K.$$

## B. Additional Experiments

### B.1. Response Cost Estimation

In BEST-Route, we estimate the incurred response cost for a given LLM and best-of-$n$ sampling strategy. The response costs can be computed by multiplying the number of output tokens by the unit output token prices (see Table 6). We use the average number of output tokens from the training split as the output length estimator for each LLM to estimate the cost. We validate that our response cost estimation is of low error and hence can be used to effectively distinguish LLMs at different cost levels for given queries (see Table 7). Specifically, the average estimation error for each query is less than $0.003 for all 8 LLMs and as low as $0.0001 for Llama-3.1-8b, Mistral-7b, and Mistral-8x7b, which demonstrates the robustness of our cost estimation.

### B.2. Performance Results Compared to Model Cascades

We compare BEST-Route with **Model Cascades** (Yue et al., 2023) to further demonstrate the effectiveness of our approach. Results are summarized in Figure 8 and Table 8. All cascading approaches incur significantly higher costs to deliver equally

| Cost Reduction (%) | Response Quality Drop (armoRM score) w.r.t. always using GPT-4o (%) | | | |
|---|---|---|---|---|
| | Cascading (exact_match) | Cascading (BLEU) | Cascading (ROUGE) | BEST-Route |
| 10 | 7.26 | 5.60 | 6.10 | **0.19** |
| 20 | 7.26 | 5.69 | 6.22 | **0.21** |
| 40 | 7.26 | 6.23 | 6.90 | **0.47** |
| 60 | 7.26 | 6.61 | 7.02 | **0.80** |

*Table 8.* Cost reduction v.s. performance drops with model cascades baselines. Performance drops are computed w.r.t. always using the reference model (GPT-4o).

good responses compared to the reference model, due to its cascading design which triggers more than one LLMs to resolve a given query. Similarly, BEST-Route outperforms all model cascades baselines by delivering higher quality responses while achieving higher cost savings. Specifically, BEST-Route achieves 60% cost reduction with 0.8% quality drop, which is up to 6.46% better than all cascading-based routers.

## C. armoRM Score - A Case Study

armoRM score (Wang et al., 2024a) is a comprehensive response quality metric that aggregates 19 evaluation perspectives, including Helpfulness, Correctness, Coherence, and Verbosity. It is constructed by collecting ground-truth ratings for each perspective, followed by normalization, de-biasing, and weighted summation into a unified score ranging from -1 to 1.

A small armoRM score increase can reflect meaningful improvements across multiple quality dimensions, as illustrated in the following example.

```
Query: Is it normal to have a fever when I'm sick?

Response 1 (armoRM = 0.112): Yes, having a fever when you're sick often indicates
    that your body is fighting off an infection or illness. Fever is a natural
    defense mechanism whereby your body's temperature increases to create an
    environment less conducive for pathogens to multiply.

Response 2 (armoRM = 0.127): Yes, it is common to have a fever when you're sick.
    A fever is your body's natural response to fighting off an infection. It
    indicates that your immune system is actively working to fight the pathogens
    causing the illness. However, if your fever is above 101°F (38.3°C) and
    persists for more than a couple of days, it's a good idea to seek medical
    advice to ensure there isn't a more serious underlying condition.
```

Both Response 1 and 2 cover the perspective that "fever is a natural defense mechanism". However, Response 2 further enriches the argument by discussing the potential danger of persisting high fever and suggests to users to seek medical advice in such cases, which could be life-critical in healthcare consultations and is missing from Response 1.

