# OpenReview forum: "BEST-Route: Adaptive LLM Routing with Test-Time Optimal Compute"
_ICML.cc/2025/Conference — ICML 2025 poster_

### Official Review · Reviewer_2U5y · 2025-03-06

**Overall Recommendation:** 3

**Summary:**

This work proposes BEST-Route, a method that combines LLM routing with test-time compute scaling.
Given a pool of LLMs and an input query, a router determines not only which LLM to route the query to, but also the number of responses to sample for it before applying Best-of-N (BoN) sampling.
Details about training the router and the proxy reward model for BoN sampling are explained.
Empirical results on various benchmarks show that the proposed method achieves a better cost-accuracy trade-off than baseline approaches.

**Update after rebuttal:** I'd like to thank the authors for their answers to my clarification questions, and will keep my positive rating unchanged.

**Claims And Evidence:**

The claims made in the submission are supported by clear and convincing evidence.

**Essential References Not Discussed:**

Not that I'm aware of.

**Experimental Designs Or Analyses:**

I have checked the experimental designs and analyses, which make sense to me. I only have a clarification question about cost calculation; see Question 1 below.

**Methods And Evaluation Criteria:**

The proposed methods and evaluation criteria make sense to me.

**Other Comments Or Suggestions:**

N/A

**Other Strengths And Weaknesses:**

N/A

**Questions For Authors:**

Some questions for clarification:

1. For the calculation of cost[(M, i)] in Algorithm 1, the number of sampled responses $i$ is multiplied only with the output length, not with the input length. Is this a typo or intentional? Are the costs in experiment results calculated in the same way? This calculation seems to rely on the assumption that the input tokens are only charged once when sampling $i$ responses for the same prompt; otherwise, it would be an underestimate of the actual cost by the proposed method.

2. What does "Random" in the legend of Figure 4 mean?

3. In Line 425 Left, the authors mention that increasing $n$ has a marginal impact on overall latency. Is this due to implementation of batch inference and parallelism?

4. Could you provide some context about the armoRM score, e.g., how significant is an increase from 0.112 to 0.126?

**Relation To Broader Scientific Literature:**

Both LLM routing and test-time scaling (via BoN sampling) have been well studied in prior works.
The key novelty in this work is a combination of both, i.e., boosting the accuracy by BoN sampling when the query is routed to a smaller model, which can still be cheaper than calling the most expensive LLM once.

**Theoretical Claims:**

There is no theoretical claim in this work.

---

> ### Author Rebuttal · Authors · 2025-04-01
>
> **Q1: For the calculation of cost[(M, i)] in Algorithm 1, the number of sampled responses i is multiplied only with the output length, not with the input length. Is this a typo or intentional? Are the costs in experiment results calculated in the same way? This calculation seems to rely on the assumption that the input tokens are only charged once when sampling i responses for the same prompt; otherwise, it would be an underestimate of the actual cost by the proposed method.**
>
> A1: Thanks for the question. This way of cost calculation is intended. Most modern LLMs support returning multiple responses at once for a given query. For example, you can set the “num_return_sequences” hyper-parameter for HuggingFace LLMs (https://huggingface.co/docs/transformers/en/main_classes/text_generation) to tune the number of independently computed returned sequences for each query. Given this, we only need to feed the prompt/input tokens once to get multiple response samples. Therefore, “input tokens are only charged once when sampling i responses for the same prompt”. We will clarify more on this in the revision.
>
> **Q2: What does "Random" in the legend of Figure 4 mean?**
>
> A2: “Random” stands for the baseline which randomly routes queries between the large and small models at different ratios. This baseline has been adopted in prior binary routing work [1, 2].
>
> **Q3: In Line 425 Left, the authors mention that increasing n has a marginal impact on overall latency. Is this due to implementation of batch inference and parallelism?**
>
> A3: Thank you for the question. This is for the same reason as explained in A1. The overall latency comprises the prefilling latency (processing the prompt tokens) and decoding latency (generating output tokens). Since we generate n responses in parallel with the same prompt (e.g., https://github.com/huggingface/transformers/blob/v4.50.0/src/transformers/generation/utils.py#L2317), increasing n only leads to modest decoding latency overhead and marginal overall latency increments (see Figure 6 in our paper). We use a batch size of 1 but larger batches may further reduce latency and are worth future exploration.
>
> **Q4: Could you provide some context about the armoRM score, e.g., how significant is an increase from 0.112 to 0.126?**
>
> A4: Thanks for the question. armoRM score [3] is a comprehensive response quality metric that aggregates 19 evaluation perspectives, including Helpfulness, Correctness, Coherence, and Verbosity. It is constructed by collecting ground-truth ratings for each perspective, followed by normalization, de-biasing, and weighted summation into a unified score ranging from -1 to 1.
>
> An increase from 0.112 to 0.126 reflects meaningful improvements across multiple quality dimensions. In our evaluation, Mistral-7B scores 0.112 on average, while GPT-4o reaches 0.126 — consistent with benchmarks like MMLU [4], where GPT-4o surpasses 85% accuracy vs. ~60% accuracy for Mistral-7B, underscoring the significance of this gap.
>
> Qualitatively, we observe that responses at 0.112 are generally helpful but limited, whereas higher-scoring responses (e.g., 0.126) offer deeper insight and more comprehensive guidance. We provide one example to illustrate this difference.
>
> “
> Query:  Is it normal to have a fever when I'm sick?
>
> Response 1 (armoRM = 0.112):
> Yes, having a fever when you're sick often indicates that your body is fighting off an infection or illness. Fever is a natural defense mechanism whereby your body's temperature increases to create an environment less conducive for pathogens to multiply.
>
> Response 2 (armoRM = 0.127):
> Yes, it is common to have a fever when you're sick. A fever is your body's natural response to fighting off an infection. It indicates that your immune system is actively working to fight the pathogens causing the illness. **However, if your fever is above 101°F (38.3°C) and persists for more than a couple of days, it's a good idea to seek medical advice to ensure there isn't a more serious underlying condition.**
> ”
>
> Both Response 1 and 2 cover the perspective that “fever is a natural defense mechanism”. However, Response 2 further enriches the argument by discussing the potential danger of persisting high fever and suggests to users to seek medical advice in such cases, which could be life-critical in healthcare consultations and is missing from Response 1.
>
> **Thank you for your time and consideration. We sincerely hope that you find our responses convincing and would consider increasing your rating.**
>
> References:
> [1] Ding, Dujian, et al. "Hybrid LLM: Cost-Efficient and Quality-Aware Query Routing." ICLR. 2024.
> [2] Ong, Isaac, et al. "RouteLLM: Learning to Route LLMs from Preference Data." ICLR. 2025.
> [3] Wang, Haoxiang, et al. "Interpretable Preferences via Multi-Objective Reward Modeling and Mixture-of-Experts." EMNLP Findings. 2024.
> [4] Hendrycks, Dan, et al. "Measuring Massive Multitask Language Understanding." ICLR. 2021.

---

### Official Review · Reviewer_HDLo · 2025-03-11

**Overall Recommendation:** 3

**Summary:**

This paper proposes a cost-effective LLM inference framework that leverages multiple small LLMs alongside a large LLM (GPT-4o). A router model is trained to (1) select which LLM to use from the candidate pool, (2) determine how many responses that LLM should generate, and (3) select the best response among them. The goal is to strike a balance between inference cost and response quality. Experimental results suggest that the proposed BEST-route method effectively achieves this balance.

**Claims And Evidence:**

The paper should avoid claiming that it achieves an optimal balance between cost and quality, as no theoretical evidence is provided to justify the optimality. A more appropriate claim would be that the method achieves a preferred or effective trade-off.

If the authors wish to make such a claim, they should justify its superiority using a well-defined metric that quantifies the trade-off between cost and quality, such as a composite metric that integrates both factors.

**Essential References Not Discussed:**

None.

**Experimental Designs Or Analyses:**

It is unclear why the experiments do not include the previous “best-of-n sampling” baselines, where multiple responses are generated and the best is selected using a reward model. These baselines are highly relevant for comparison, as they also aim to balance cost and quality. The authors should either include such baselines or provide a clear justification for their omission.

**Methods And Evaluation Criteria:**

The paper evaluates response quality solely using armoRM, which is also the training objective for the method. This introduces potential bias and may not fully capture response quality. The reviewer recommends incorporating additional evaluation metrics, such as BLEU, ROUGE, or human evaluation, to provide a more comprehensive assessment.

**Other Comments Or Suggestions:**

In section 4.2, the "They key intuition" should be "The key intuition".

**Other Strengths And Weaknesses:**

The paper does not sufficiently describe the architecture of the proxy reward model
$R_{proxy}(s)$. While the cost-efficient multi-head router is stated to use a BERT-style backbone, no analogous detail is given for the reward model. This information is crucial for reproducibility and understanding the model’s capacity.

**Questions For Authors:**

The paper does not clarify the output range of the proxy reward model
$R_{proxy}(s)$. Is the score normalized to lie within [0,1]? Knowing the score range is important for interpreting the reward model’s outputs and understanding how it influences selection (e.g., the $L_{rank}$ in equation (1)).

**Relation To Broader Scientific Literature:**

The paper reveals that there is potential to actively select between LLMs with different costs. The trade-off between cost and performance can be vital.

**Theoretical Claims:**

No theoretical analysis is included.

---

> ### Author Rebuttal · Authors · 2025-04-01
>
> We thank the reviewer for thoughtful comments. Due to space limits, we have paraphrased some questions to save space while preserving the original intent.
>
> **Q1: The claim of “optimal balance” between cost and quality is too strong without theoretical justification. A more appropriate phrasing would be “preferred” or “effective” trade-off, unless supported by a formal composite metric.**
>
> A1: Thanks for the comment. This work aims to effectively uplift the performance-cost trade-offs achieved by routing techniques. We will revise the claim as “BEST-Route achieves an effective trade-off between cost and quality”, as suggested.
>
> **Q2: The exclusive use of armoRM (also used for training) may introduce bias. Additional metrics like BLEU, ROUGE, or human evaluation are recommended for a more comprehensive assessment.**
>
> A2: Thanks for the comment. We now report BLEU and ROUGE scores, as shown in the tables below. Since human evaluation is expensive and unscalable, we leave it in future work. We observe that BEST-Route consistently achieves better trade-offs than all baselines (e.g., N-label routing has up to 31.7% BLEU drop at 60% cost reduction, vs. only 18.07% drop for BEST-Route).
>
> |  | Response Quality Drop (BLEU score) | w.r.t. always using GPT-4o (%) |
> |:---:|:---:|:---:|
> | Cost Reduction (%) |     N-label   | BEST-Route |
> | 10 | 6.57 | **3.61** |
> | 20 | 13.13 | **6.07** |
> | 40 | 25.80 | **12.76** |
> | 60 | 31.70 | **18.07** |
> | N-class | N-class achieves 0.2% cost reduction | with 0.7% quality drop. |
> | Clustering | Clustering achieves 0% cost reduction | with 0% quality drop. |
>
> |  | Response Quality Drop (ROUGE score) | w.r.t. always using GPT-4o (%) |
> |:---:|:---:|:---:|
> | Cost Reduction (%) |     N-label   | BEST-Route |
> | 10 | 6.10 | **3.88** |
> | 20 | 11.65 | **7.27** |
> | 40 | 22.26 | **15.78** |
> | 60 | 27.62 | **21.97** |
> | N-class | N-class achieves 0.2% cost reduction | with 0.9% quality drop. |
> | Clustering | Clustering achieves 0% cost reduction | with 0% quality drop. |
>
> A detailed visualization is provided in Figure 2 and 3, via [link](https://github.com/BEST-Route2025/BEST-Route/blob/main/README.md).
>
> **Q3: It is unclear why the experiments do not include the previous “best-of-n sampling” baselines, where multiple responses are generated and the best is selected using a reward model. These baselines are highly relevant for comparison, and should be included or clearly justified.**
>
> A3: Thanks for the comment. We have implemented the best-of-n sampling baseline and report the performance in Figure 1, via [link](https://github.com/BEST-Route2025/BEST-Route/blob/main/README.md). Best-of-n offers fixed trade-offs for each model and n pair and often yields lower-quality responses (e.g., 4.9% quality drop for Phi-3-mini, 1.1% for LLaMA-3.1-8B at n=5). In contrast, BEST-Route offers flexible trade-offs, achieving 20% cost reduction with only 0.21% quality drop, and 40% cost reduction with 0.47% drop, with the max sampling number n=5.
>
> **Q4: The architecture of the proxy reward model R_proxy(s) is unclear. While the cost-efficient multi-head router is stated to use a BERT-style backbone, no analogous detail is given for the proxy reward model.**
>
> A4: In Lines 283-286, we mentioned “the proxy reward model is fine-tuned from OpenAssistant RM, a DeBERTa-v3-large model (300M)”. Specifically, the proxy reward model is a DeBERTa model [1] which improves BERT with enhanced mask decoder and disentangled attention. In our evaluation, we observe that the proxy reward model is cost-efficient and incurs negligible overhead (see Figure 6 in our paper).
>
> **Q5: In section 4.2, the "They key intuition" should be "The key intuition".**
>
> A5: Thank you. We will fix this in the revision.
>
> **Q6: The output range of R_proxy(s) is unclear. Clarifying whether scores are normalized (e.g., within [0,1]) is important for interpreting outputs and their role in selection (e.g., the L_rank in equation (1)).**
>
> A6: Thanks for the comment. In our paper, we train a proxy reward model to “preserve the ranking of responses” and take the output logits as the proxy reward scores R_proxy(s), which ranges from (-∞, +∞). A higher proxy score indicates a better response quality. In our evaluation, proxy reward scores range from -12.25 to 12.1875, as detailed in Figure 4, accessible via [link](https://github.com/BEST-Route2025/BEST-Route/blob/main/README.md).
>
> In equation (1), we construct training pairs in the form of (good response s, bad response s’) and train the proxy reward model to preserve the ranking of responses by minimizing the negative log likelihood loss L_rank, which takes on lower values as the proxy reward score difference, R_proxy(s) - R_proxy(s’), gets larger .
>
> **Thank you for your time and consideration. We sincerely hope that you find our responses convincing and would consider increasing your rating.**
>
> References:
> [1] He, Pengcheng, et al. "DEBERTA: DECODING-ENHANCED BERT WITH DISENTANGLED ATTENTION." ICLR 2021.

---

### Official Review · Reviewer_Yhv8 · 2025-03-13

**Overall Recommendation:** 4

**Summary:**

This paper proposes BEST-Route, combining two different research areas: model selection (through routing) with adaptive allocation of test-time compute (e.g., best-of-n sampling). Their framework dynamically selects a model and the optimal number of responses to sample based on query difficulty and quality thresholds. Experimental results show up to 60% cost savings with less than 1% performance degradation compared to always using a large model.

**Claims And Evidence:**

- The main claim, “Experiments on real-world datasets demonstrate that our method reduces costs by up to 60% with less than 1% performance drop”, is validated empirically on a dataset compiled from existing sources (Table 3).

- The claim “surpassing prior routing approaches and setting a new standard for efficient LLM service deployment” seems too strong. While the method shows clear improvements over prior routing approaches, it would benefit from direct comparisons with other efficiency techniques, such as speculative decoding, as mentioned in the related work.

**Essential References Not Discussed:**

See below.

**Experimental Designs Or Analyses:**

See the comments above.

**Methods And Evaluation Criteria:**

The methods and evaluation criteria seem reasonable. However,  if I understood correctly, the router is trained on data that is very similar to the validation and test data (since all splits are sampled from the dataset that they collected). While this is acknowledged as a limitation, additional discussion is needed on how well BEST-Route generalizes to unseen datasets. Ideally, testing on another dataset would strengthen the results. This is the main weakness of the paper, in my opinion.

**Other Comments Or Suggestions:**

- “The rising costs of LLM inference have”. I get your point but it would be good to provide some examples (e.g., self-consistency, reranking, etc.)

- “3) Model cascades ( e.g. Chen et al. (2023)) where the query passes through the models sequentially, from the cheapest to the most costly, until a satisfactory response is obtained.”- Not always “until a satisfactory response is obtained”… most cases consider a fixed number of models in the cascade

- “Prior query routing approaches generate only one response from the selected model and a single response from a small (inexpensive) model was often not good enough to beat a response from a large (expensive) model due to which they end up overusing the large model and missing out on potential cost savings.” and “, small models continue to come up short in terms of response quality when compared to the largest, most powerful models”. While I agree with this, there are cases when this is not the case. For instance, check Table 1 of Farinhas et al. (2025). Even though this is for model cascading and specific for machine translation, I think it’s worth mentioning in Section 2.

- Typo in L048-049: “development innovative solutions”


References:

Farinhas et al., 2025. Translate Smart, not Hard: Cascaded Translation Systems with Quality-Aware Deferral


**Update after the rebuttal**: I increased my score to 4.

**Other Strengths And Weaknesses:**

Strengths:  The motivation is well explained, and the topic is important, especially now with the trend of building larger and larger models. I particularly appreciate Section 3.1. Also, combining test time computing with routing makes a lot of sense and has not been explored before (to my knowledge).

**Questions For Authors:**

NA

**Relation To Broader Scientific Literature:**

The paper combines research on query routing with test-time compute and cites related work properly.

**Theoretical Claims:**

NA

---

> ### Author Rebuttal · Authors · 2025-04-01
>
> **Q1: The claim “surpassing prior routing approaches and setting a new standard for efficient LLM service deployment” seems too strong. While the method shows clear improvements over prior routing approaches, it would benefit from direct comparisons with other efficiency techniques, such as speculative decoding.**
>
> A1: Thanks for the comment. We will revise the claim to emphasize that our method significantly improves upon prior routing techniques, contributing toward more efficient LLM service deployment. It is worth noting that our approach is orthogonal and complementary to efficiency techniques like speculative decoding [1], which accelerates decoding for expensive models. In contrast, BEST-Route reduces cost by intelligently assigning “easy” queries to small models while maintaining high performance. A hybrid system could combine the advantages of both techniques and could first route queries (via BEST-Route) between cheap and expensive LLMs, and then apply speculative decoding if the expensive model is selected to yield further efficiency gains.
>
> **Q2: Additional discussion is needed on how well BEST-Route generalizes to unseen datasets. Ideally, testing on another dataset would strengthen the results. This is the main weakness of the paper, in my opinion.**
>
> A2: Thanks for this insightful comment. We evaluate the trained routers on the OOD dataset, MT-Bench [2], and observe that BEST-Route achieves strong performance under data distribution shifts (e.g., 60% cost reduction with only 1.59% quality drop). Due to the space limit, please refer to A2 for Reviewer ewCv for details.
>
> **Q3: “The rising costs of LLM inference have”. I get your point but it would be good to provide some examples (e.g., self-consistency, reranking, etc.)**
>
> A3: Thanks for the comment. We will revise the introduction by including examples on “the rising costs of LLM inference”, such as self-consistency [3], which samples multiple reasoning paths and selects the most consistent answer at increased cost, and reranking [4], which generates multiple candidates and uses a re-ranker to select the ones that could lead to better final results at increased inference costs.​
>
> **Q4: “3) Model cascades ( e.g. Chen et al. (2023)) where the query passes through the models sequentially, from the cheapest to the most costly, until a satisfactory response is obtained.”- Not always “until a satisfactory response is obtained”… most cases consider a fixed number of models in the cascade**
>
> A4: Thank you for the comment. We will revise to clarify that, most cascade approaches consider sequentially executing models, until either a satisfactory response is obtained or a pre-defined max number of models of the cascade is reached [5].
>
> **Q5: “... a single response from a small (inexpensive) model was often not good enough to beat a response from a large (expensive) model …” and “small models continue to come up short in terms of response quality when compared to the largest, most powerful models”. While I agree with this, there are cases when this is not the case. For instance, check Table 1 of Farinhas et al. (2025). Even though this is for model cascading and specific for machine translation, I think it’s worth mentioning in Section 2.**
>
> A5: Thank you for pointing this out. We agree and we have independently verified this behaviour in our evaluation. While small models underperform large ones on average, specific cases exist where they perform comparably or even better. Farinhas et al. (2025) observed that though Tower-v2 7B is inferior to its large counterpart Tower-v2 70B on average, it can outperform the large model on 32% cases. This observation motivates routing queries between LLMs to not only save costs but also improve performance. Table 2 in our paper empirically supports this intuition by showing cases where routing achieves significant cost reduction (e.g., 20%) as well as performance improvements (e.g., 0.5%). We will clarify more on this observation in our revision.
>
> **Q6: Typo in L048-049: “development innovative solutions”**
>
> A6: Thank you. We will fix this in the revision.
>
> **Thank you for your time and consideration. We sincerely hope that you find our responses convincing and would consider increasing your rating.**
>
> References:
> [1] Leviathan, Yaniv, et al. "Fast inference from transformers via speculative decoding." ICML 2023.
> [2] Zheng, Lianmin, et al. "Judging llm-as-a-judge with mt-bench and chatbot arena." NeurIPS 2023.
> [3] Wang, Xuezhi, et al. "Self-consistency improves chain of thought reasoning in language models." arXiv 2022.
> [4] Chuang, Yung-Sung, et al. "Expand, Rerank, and Retrieve: Query Reranking for Open-Domain Question Answering." ACL Findings 2023.
> [5] Chen, Lingjiao, et al. "Frugalgpt: How to use large language models while reducing cost and improving performance." arXiv 2023.
> [6] Farinhas, António, et al. "Translate Smart, not Hard: Cascaded Translation Systems with Quality-Aware Deferral." arXiv 2025.

---

### Official Review · Reviewer_ewCv · 2025-03-17

**Overall Recommendation:** 4

**Summary:**

The paper introduces BEST-Route, an adaptive routing framework designed to optimize inference efficiency and response quality in large language models (LLMs). The framework dynamically selects an appropriate LLM and determines the optimal number of responses to sample (best-of-n sampling) based on the estimated difficulty of individual queries. It builds on the observation that combining multiple responses from smaller, more cost-effective models, along with a lightweight proxy reward model to select the best response, significantly enhances quality while often being more economical than consistently using larger, costly models.

Main contributions of the paper are:
* Introduces a dynamic multi-head router that adaptively assesses query difficulty to efficiently allocate computational resources, optimizing the trade-off between inference cost and response accuracy.

* Employs best-of-n sampling to produce multiple candidate responses from smaller models, selecting the highest-quality response through an efficiently trained proxy reward model.

* Empirical evaluations demonstrate that BEST-Route significantly improves inference efficiency, achieving up to 60% cost reduction with less than a 1% decrease in response quality compared to consistently utilizing a state-of-the-art reference model (GPT-4o).

**Claims And Evidence:**

Supported claims:
* The paper demonstrates the effectiveness of its multi-head router architecture, supported by comprehensive experiments across a diverse set of real-world tasks.
* The paper demonstrates the performance gains from utilizing best-of-n sampling on smaller models through detailed quantitative results.
* Detailed analyses against baseline methods (N-label, N-class, clustering, and cascade methods) support the claim that BEST-Route achieves significant inference cost reductions with minimal impact on response quality

Problematic claims:
* The authors present clear results across diverse tasks, yet the evaluation is primarily conducted on datasets curated specifically for this paper. Additional evidence showing performance consistency on independently curated datasets or real-world deployments could further strengthen the generalizability of their approach.
* The submission briefly acknowledges potential sensitivity to data drift but does not provide empirical evidence or in-depth analysis regarding the robustness of BEST-Route under changing data distributions. Addressing this gap through empirical evaluation or detailed discussion would be beneficial for supporting practical deployment scenarios.

**Essential References Not Discussed:**

NA

**Experimental Designs Or Analyses:**

Please see methods section above

**Methods And Evaluation Criteria:**

* The proposed BEST-Route framework introduces a novel multi-head routing architecture, integrating best-of-n sampling for smaller models
* The proposed benchmark dataset covers a range of important application scenarios (e.g., question answering, coding, and safety evaluation)
* However, although the dataset provides a valuable evaluation framework across multiple tasks, it doesn't show the generalizability of the approach especially with data drift in real-world cases.

**Other Comments Or Suggestions:**

Update: The provided rebuttal strengthens the paper so I increased my score to 4.

**Other Strengths And Weaknesses:**

The paper is very well written and easy to follow

**Questions For Authors:**

NA

**Relation To Broader Scientific Literature:**

The paper contributes to the active area of cost-efficient LLM inference by combining adaptive routing techniques with test-time optimal compute strategies.

**Theoretical Claims:**

NA

---

> ### Author Rebuttal · Authors · 2025-04-01
>
> We thank the reviewer for the thoughtful and constructive comments. Several comments pertain to the generalizability of BEST-Route, and we address them jointly below for clarity.
>
> **Q1: The authors present clear results across diverse tasks, yet the evaluation is primarily conducted on datasets curated specifically for this paper. Additional evidence showing performance consistency on independently curated datasets or real-world deployments could further strengthen the generalizability of their approach.**
>
> A1: Thanks for this comment. We would like to clarify that our dataset is a random sample of multiple public benchmarks such as CodeUltraFeedback [1] and BeaverTails [2], which are independently curated by other parties. That said, to further evaluate generalizability, we test BEST-Route on the out-of-distribution (OOD) benchmark MT-Bench [3], a widely used dataset covering writing, reasoning, fact extraction, and roleplay. Please see A2 below for details.
>
> **Q2: The submission briefly acknowledges potential sensitivity to data drift but does not provide empirical evidence or in-depth analysis regarding the robustness of BEST-Route under changing data distributions. Addressing this gap through empirical evaluation or detailed discussion would be beneficial for supporting practical deployment scenarios.
> Q3: However, although the dataset provides a valuable evaluation framework across multiple tasks, it doesn't show the generalizability of the approach especially with data drift in real-world cases.**
>
> A2: Thanks for highlighting this. We evaluate BEST-Route on the OOD dataset MT-Bench [3]. As shown in the table below, BEST-Route consistently outperforms all baselines. For example, it achieves 60% cost reduction with only a 1.59% performance drop – up to 4.3% better than the strongest baseline. In contrast, N-class and clustering-based routing often default to using GPT-4o, yielding minimal cost savings, while N-label routing suffers notable quality drops especially at high cost reduction rates. These results demonstrate BEST-Route's robustness under distribution shifts.
>
> |  | Response Quality Drop (armoRM score) | w.r.t. always using GPT-4o (%) |
> |:---:|:---:|:---:|
> | Cost Reduction (%) |     N-label   | BEST-Route |
> | 10 | 0.88 | **0.25** |
> | 20 | 2.29 | **0.43** |
> | 40 | 4.41 | **1.56** |
> | 60 | 5.89 | **1.59** |
> | N-class | N-class achieves 0% cost reduction  | with 0% quality drop. |
> | Clustering | Clustering achieves 0% cost reduction | with 0% quality drop. |
>
> A detailed visualization of quality-v.s.-cost trade-offs achieved by different methods on MT-Bench is provided in Figure 5, accessible via the anonymous [link](https://github.com/BEST-Route2025/BEST-Route/blob/main/README.md#figure-5-routing-performance-results-on-the-ood-dataset-mt-bench).
>
> **Thank you for your time and consideration. We sincerely hope that you find our responses convincing and would consider increasing your rating.**
>
> References:
> [1] Weyssow, Martin, et al. "Codeultrafeedback: An llm-as-a-judge dataset for aligning large language models to coding preferences." arXiv preprint arXiv:2403.09032 (2024).
> [2] Ji, Jiaming, et al. "Beavertails: Towards improved safety alignment of llm via a human-preference dataset." Advances in Neural Information Processing Systems 36 (2023): 24678-24704.
> [3] Zheng, Lianmin, et al. "Judging llm-as-a-judge with mt-bench and chatbot arena." Advances in Neural Information Processing Systems 36 (2023): 46595-46623.

---

### Decision · Program_Chairs · 2025-05-01

**Decision:**

Accept (poster)

**Comment:**

This paper works on query routing for LLMs, and proposes a method called BEST-Route, which optimally balances cost and quality through: 1) multi-headed router that dynamically assesses the query difficulty for model selection and 2) a test-time optimal compute strategy that uses best-of-n sampling. The rebuttal addressed the reviewers' initial concerns, and all reviewers recommend acceptance after the rebuttal phase. The ideas are novel and interesting, and the motivation is clear. I thus recommend acceptance.